# Species-Specific Deoxyribonucleic Acid (DNA) Identification of Bovine in Cultured Meat Serum for *halal* Status

**DOI:** 10.3390/foods11203235

**Published:** 2022-10-17

**Authors:** Mohd Izhar Ariff Mohd Kashim, Alia Aryssa Abdul Haris, Nur Asmadayana Hasim, Sahilah Abd Mutalib, Nurina Anuar

**Affiliations:** 1Research Centre of Sharia, Faculty of Islamic Studies, Universiti Kebangsaan Malaysia, Bangi 43600, Selangor, Malaysia; 2Institute of Islam Hadhari, Universiti Kebangsaan Malaysia, Bangi 43600, Selangor, Malaysia; 3Department of Food Science, Faculty of Science and Technology, Universiti Kebangsaan Malaysia, Bangi 43600, Selangor, Malaysia; 4Innovation Centre for Confectionery Technology (MANIS), Faculty of Science and Technology, Universiti Kebangsaan Malaysia, Bangi 43600, Selangor, Malaysia; 5Department of Chemical and Process Engineering, Faculty of Engineering and Built Environment, Universiti Kebangsaan Malaysia, Bangi 43600, Selangor, Malaysia

**Keywords:** cultured meat, polymerase chain reaction (PCR), bovine serum, *halal*

## Abstract

Meat culturing technology goes beyond laboratory research and materialises in the market. Nonetheless, this technology has raised concerns among Muslim consumers worldwide due to its medium, especially foetal bovine serum (FBS), which originates from blood. Thus, the aim of this research was to determine the *halal* status of cultured meat by detecting species-specific DNA of bovine serum as one of the media used during meat production. Polymerase chain reaction (PCR) analysis was conducted by targeting mitochondrial cytochrome oxidase II (COII) gene sequences, producing a 165 bp amplicon. The sequences of the primers used were Bovine-F, 5′-CAT CAT AGC AAT TGC CAT AGT CC-3′ and Bovine-R, 5′-GTA CTA GTA GTA TTA GAG CTA GAA TTA G-3′. DNA extraction was conducted using a QIAGEN Blood and Tissue™ commercial kit. The presence study also included a literature review on the *Istihalah* (transformation) concept in order to determine the *halal* status of cultured meat. The results revealed that bovine DNA was detected in all samples tested using PCR analysis. Therefore, *Istihalah tammah* (perfect transformation) does not occur due to the ability of PCR analysis to detect bovine DNA in FBS and is prohibited according to Shariah law.

## 1. Introduction

The presence of modern food products is increasing in the market owing to their ease of production, cost-effectiveness and high demand in urban society [1]. Modern food is defined as food produced with advanced food science and technology in the modern era [2]. Modern food products exposed are produced using raw material sources mixed with gelatine; fat; blood; alcohol; parts of unslaughtered animals; carcasses; and prohibited animals, such as pigs [3]. This causes concern among consumers, especially Muslims, in choosing modern food products that are *halal* (permitted). The word ‘*halal*’ is derived from and Arabic word that means allowed or permitted [1]. A *halal* food designation indicates that the food is allowed to be eaten by Muslims [1]. Recently, Singapore announced that the largest cultured meat facility in Asia is scheduled open in 2023 [4]. This news has induced anxiety among Muslim consumers, especially in Malaysia, with a 60% Muslim population and located near Singapore [5].

According to Sirini and colleagues [6], recent studies have shown that there is consumer demand for healthier and functional formulations of meat products. Generally, cultured meat offers eclectic benefits and resolves financial, animal welfare, ethics, resource scarcity and general health issues [7,8]. By definition, ‘cultured meat’ is in vitro, synthetic or lab-grown meat produced in a bioreactor through tissue engineering technology [8]. Bhat et al. [9] noted that cultured meat is produced outside the animal body, i.e., through cell production with specialised stem cells. Stem cells and tissues are then placed in a medium appropriate for their growth and maturation into muscle fibre, which is the essential component of meat. The medium used should also contain all the nutrients and substrates required by cells and tissues or stem cells to multiply and mature [9].

In cultured meat production, foetal bovine serum (FBS) is often used as a medium. The culture medium contains amino acids, vitamins, inorganic salts, glucose, growth factors (hormones) and other nutrients needed for cell growth [10]. FBS is a fraction of clotted blood fluid from a bovine fetus, cell-free fibrin and clotting factors, with high contents of nutrients and macromolecules essential for cell growth. In addition, FBS is a universal supplement containing 200–400 distinct proteins and thousands of small-molecule metabolites, such as amino acids, sugars, lipids and hormones, in unknown concentrations. Hormones, vitamins, transport proteins, trace minerals and growth factors are vital for cell proliferation and maintenance [11]. Jochems et al. [12] reported that FBS is collected from the blood of unborn calves during the last two trimesters of pregnancy, mostly during the slaughter of pregnant cows. Other types of serum include newborn calf serum and horse serum. Growth factors are currently extracted from cattle foetuses, but they are difficult to obtain in large quantities, and purification methods are complex and expensive [10].

New technology always brings new challenges and opportunities to the religious community [13]. For Muslim consumers, cultured meat technology raises doubts due to the use of serum in meat cell culturing [10,14]. Serum is a byproduct of blood obtained by a coagulation and centrifugation process to remove the clot and blood cells. According to Malaysian Standard MS1500: 2019 *halal* Food, blood is classified as *al-mutawassitah najs* (intermediate impurity) and is haram for consumption [15,16]. Plasma and serum are categorised as *al-mutawassitah najs* because they are obtained from blood. However, plasma and serum can be used if the food/final product goes through an *istihalah* (transformation) process, which is a terminological process that transforms or converts a substance (*najs*/impurity) through changes in its composition and properties [17]. The detection of species-specific DNA in blood plasma was reported by Shahimi et al. [18] in chicken meatballs made with fresh and commercial blood plasma. Essentially, it involves detection of agents of change either naturally or unnaturally through synthetic processes or human intervention. In the discipline of fiqh, this theory is part of an alternative purification mechanism called *Istihalah* (transformation)*. Istihalah* is a process that allows an element or a component to be used if it changes through a natural process. *Istihalah tammah* or perfect transformation is the process of *najs* conversion to a new product that differs entirely from the original product. The aim of this process is to purify *najs* or materials that come into contact with *najs*.

Perfect transformation is similar to the concepts of ethanol change to acetic acid. Ethanol is prohibited for consumption if it is derived from a fermentation process and *halal* if it is derived from other processes. Acetic acid is *halal* and derived from the oxidative fermentation of acetic acid bacteria, such as *Acetobacter* and *Gluconobacter* genera [19]. The concept of ethanol-to-acetic transformation involves the idea of existence or presence; therefore, PCR is used to determine the presence of serum. According to Shariah law, cattle are among *halal* animals to be consumed when they are slaughtered [1]. If bovine DNA is not detected in bovine serum, this indicates that the *istihalah* process has taken place, and there is no doubt about using bovine serum as a medium. On the other hand, if bovine DNA can be detected in the tested serum medium, this indicates that the serum was not transformed sufficiently, such as by ethanol to acetic acid. With respect to serum transformation from blood, the colour changes from red to a yellow, transparent liquid. The addition of DNase in the serum will eliminate the DNA detection.

Several methods for species identification of meat are available, including DNA [20,21,22], protein [23,24] and fat-based analysis [25]. Species-specific identification of animals based on DNA analysis, such as PCR-based assays, have been applied to detect samples of meat, milks products, gelatine and blood plasma [18,26,27,28]. Animal DNA detection has not been reported in serum to date. Mid-infrared (MIR) spectroscopy and enzyme-linked immunosorbent assays have been used for protein detection [23,24]. Fourier transform infrared (FTIR) spectroscopy was used to evaluate offal in beef patties [25]. In the present study, DNA-based PCR was chosen due to its frequent use to evaluate authenticity, fraud and adulteration of food components. This technique is easy, simple, sensitive and relatively low-cost. The aim of this study is to determine the *halal* status of cultured meat using FBS as a medium, which can be detected by PCR analysis. Hence, this study serves as a reference for stakeholders in the food industry, as well as Muslim consumers. PCR is an easy, fast and reproducible technique to identify species-specific animal DNA in the event of contamination by blood sources [29]. In this study, we compared DNA detection between bovine serum and bovine blood plasma. PCR detection can determine whether an *Istihalah tammah* (perfect transformation) process has occurred, indicating the *halal* status of cultured meat.

## 2. Materials and Methods

### 2.1. Samples for Analysis

Blood plasma and bovine serum (*Bos taurus*) were commercially obtained from Sigma-Aldrich. Blood plasma was received powdered form, whereas the serum was obtained in liquid form. The powdered form of blood plasma was diluted with 10 mL of distilled water. Commercial bovine genomic DNA (Novagen^®^, Darmstadt, Germany) was used as a positive control.

### 2.2. DNA Extraction

A QIAGEN Blood and Tissue™ commercial kit (Hilden, Germany) was used for DNA extraction from serum and plasma. The following materials were used for each extraction: 360 µL food lysis buffer (ATL buffer), 20 µL proteinase K, 200 µL AL buffer, 200 µL 100% (*v*/*v*) ethanol, 500 µL AW1 buffer, 500 µL AW2 buffer and 100 µL AE buffer (QIAGEN, Hilden, Germany). Sample volumes of 300 µL and 500 µL were used for extraction. Extracted DNA was stored at −20 °C until further analysis. All DNA samples were extracted in duplicate from each source.

### 2.3. Oligonucleotide Primers

A pair of species-specific primers were used in each PCR assay for bovine detection. Bovine-F/Bovine-R primers were used to detect bovine DNA in the following sequence: Bovine-F, 5′-CAT CAT AGC AAT TGC CAT AGT CC-3′ and Bovine-R, 5′-GTA CTA GTA GTA TTA GAG CTA GAA TTA G-3′ [30]. These primers targeted the mitochondrial DNA of the cytochrome oxidase II (COII) gene sequence and produced an amplicon size of 165 bp. All primers were supplied by Apical Scientific (Seri Kembangan, Selangor, Malaysia).

### 2.4. Polymerase Chain Reaction (PCR)

The PCR simplex amplification technique using COII primers targeting mtDNA was performed at a final volume of 50 μL; the volumes of each PCR mixtures are shown in Table 1. A mastercycler^®^ gradient thermal cycler (Eppendorf, Westbury, NY, USA) was used to run the PCR with a temperature program as stated in Table 2. The detection limit reported for the analysis was as low as 0.1 ng DNA [30].

The amplification products were electrophoresed through a 2.5% (*w*/*v*) agarose gel in a 1 × TAE buffer (40 mM Tris-acetate, 1 mM EDTA, pH 8.0) at 100 V for 45 min and pre-stained with Maestrosafe^TM^ nucleic acid (V-BioScience, Kuala Lumpur, Malaysia) [31]. Commercial bovine DNA NovaGen^®^ (Merck, Darmstadt, Germany) was used as a positive control. All of the agarose gel electrophoreses of the PCR product used a GeneRuler^TM^ 100-bp DNA ladder (Fermentas, Vilnius, Lithuania) as a molecular size marker and were visualized using a UV gel documentation system (Syngene, Cambridge, UK).

## 3. Results and Discussion

This study was conducted to detect animal DNA in serum and blood plasma. The results can be used by stakeholders with respect to decisions concerning cultured meat consumption. The issue of law (hukm) is relevant to cultured serum media when the use of foetal bovine serum (FBS) is unavoidable laboratory meat culture processes. Carlo [12] claimed that serum is a solution without platelets and clotting factors, i.e., fibrinogen cells. FBS is directly involved in the production process of cultured meat, affecting purity of the meat produced. FBS is a major growth factor in the production of cultured meat. Islam prohibits the intake of animal blood, and it is believed to contain a variety of potentially harmful microorganisms and metabolic products and toxins [1,32]. The mixture of FBS components is believed to contain many unknown chemical constituents, as well as contaminants, such as endotoxins, mycoplasms, viruses and prion proteins [33]. Therefore, the use of blood serum can be detrimental to health, for example, causing allergies caused by blood proteins. Flowing blood is classified as an element of *najis*, which is agreed (ijma’) by scholars to be prohibited [34].

In the present study, bovine blood plasma and serum showed bands at the 165 bp position, targeting the cytochrome oxidase II (COII) gene sequence for bovine species. No statistical analysis was applied, samples were examined in duplicates (D) for 300 µL and triplicates for 500 µL blood plasma samples. Figure 1 shows the amplicons for 500 µL bovine blood plasma samples, labelled as D1, D2 and D3, and 300 µL bovine blood plasma samples, labelled as D4 and D5.

The positive control was labelled as PC and produced 165 bp amplicons. Based on agarose gel observation, all blood plasma samples were in line with the PC, indicating a similar molecular size. For the negative control (NC), no DNA band was observed due to the absence of DNA. As shown in Figure 1, lanes 1–3 were brighter than lanes 4 and 5, indicating that the DNA intensity was aligned with the concentration of DNA. The higher the volume of extracted DNA, the higher the concentration of obtained DNA and therefore increased intensity. This finding is in line with a study by Shahimi et al. [18], who reported that thicker bands showed more DNA was available and amplified by the primers.

Bovine serum amplicons with a volume of 500 µL (D1, D2 and D3) a volume of 300 µL (D4, D5 and D6) are shown in Figure 2. All amplicons showed at 165 bp with similar intensities. Sample volume did not seem to have an effect on blood plasma DNA. The intensities of all samples were identical, although the DNA was extracted from different volumes. The appearance of serum is similar to that of blood plasma, appearing as a yellow transparent solution that does not contain fibrinogen, which prevents blood clotting when used as a medium for the proliferation of cultured meat. The intensity of the bands in blood plasma was greater than that in serum. The serum may have a lower concentration of DNA than plasma after having undergone fibrinogen removal. Blood serum PCR analysis was not conducted on animal DNA.

The application of mitochondrial DNA (mtDNA) COII as targeted gene was previously reported by Corona et al. [30]. Similar results are presented here; both blood plasma and serum showed bands at the 165 bp amplicon. mtDNA using COII as targeted gene results in a high probability of positive results from samples subjected to high temperatures during processing [35,36]. Azli et al. [34] also reported the detection of bovine DNA from ghee treated at high temperatures. PCR analysis has the ability to detect DNA in food products [37,38,39] associated with blood plasma as ingredients, despite having undergone processing, such as thermal and high-pressure treatment [31,40,41].

## 4. Conclusions

In the present study, PCR analysis was used to detect animal DNA in serum. The use of any component derived from blood is prohibited in Islam. Cultured meat is an innovative meat product that has the potential to materialised in a near future as an alternative to conventional meat. However, concerns with respect to the technology of cultured meat include the use of serum as a component of growth media. Muslim consumers are only permitted to consume foods that are in compliance with Shariah laws. As demonstrated in the present study, serum does not undergo perfect transformation or *Istihalah tammah*; based on the detection of species-specific animal DNA in serum, cultured meat is prohibited for consumption.

## Figures and Tables

**Figure 1 foods-11-03235-f001:**
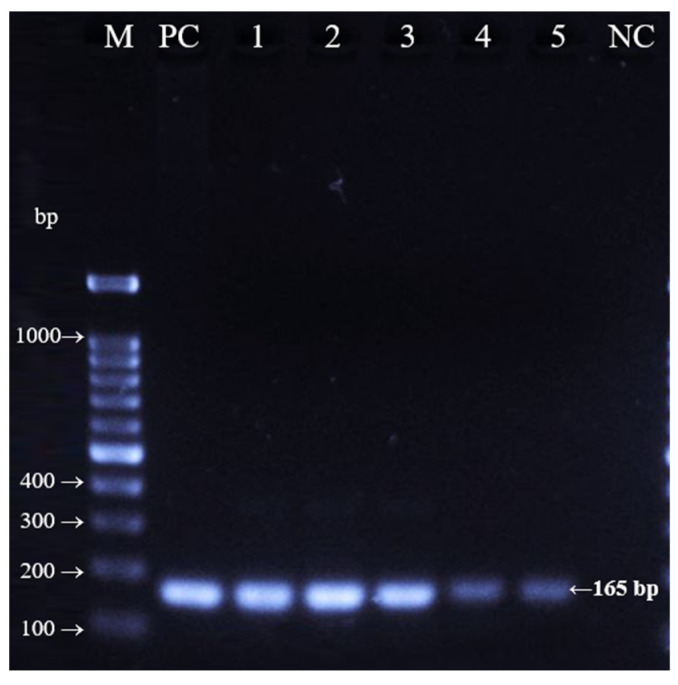
PCR amplification results for bovine blood plasma samples. M Lane: 100 bp marker; PC lane: positive control (165 bp); lane 1: D1; lane 2: D2; lane 3: D3; lane 4: D4; lane 5: D5; NC: negative control.

**Figure 2 foods-11-03235-f002:**
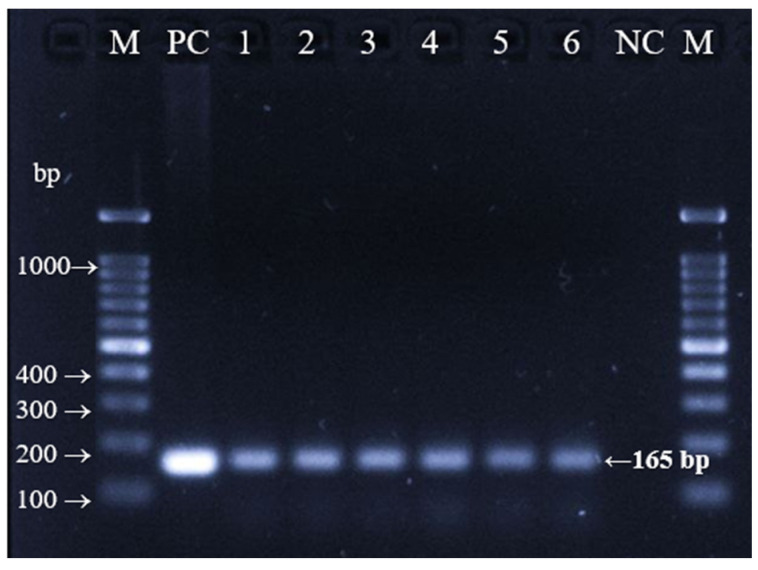
PCR amplification results for bovine serum samples. M Lane: 100 bp marker; PC lane: positive control (165 bp); lane 1: D1; lane 2: D2; lane 3: D3: lane 4: D4; lane 5: D5; lane 6: D6; NC: negative control.

**Table 1 foods-11-03235-t001:** Volume of PCR mixture.

PCR Mixture Material (µL)	Sample	Positive Control	Negative Control
DreamTaq parent mix™ green PCR (2×)	25	25	25
Primer (front)	1.0	1.0	1.0
Primer (rear)	1.0	1.0	1.0
DNA template	2	–	–
DNA samples	–	2	–
Nuclease-free water (NFW)	23	23	23
**Total volume (µL)**	**52.0**	**52.0**	**50.0**

**Table 2 foods-11-03235-t002:** PCR program for bovine DNA detection.

Primer	Method	Temperature (°C)/Time (min)	Reference
Bovine-F/Bovine-R	Initial denaturation	95/2	Corona et al. [30]
	Denaturation	94/1	
	Annealing	55/1	
	Elongation	72/2	
	Annealing	55/1	
	Elongation	72/2	
	Final elongation	72/10	

## Data Availability

The datasets generated for this study are available on request to the corresponding author.

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
