# Peer review of "Species-Specific Deoxyribonucleic Acid (DNA) Identification of Bovine in Cultured Meat Serum for halal Status"

_foods, 2022, doi:10.3390/foods11203235_

Round 1

Reviewer 1 Report

This manuscript describes a PCR method for detection of animal DNA in serum. PCR has been a mature technology in the field of nucleic acid detection. That is, authors are harnessing readily available technologies to detect the DNA.
The main concern for this manuscript is the novelty. However, PCR is the most conventional method to analysis DNA. Another concern is the experiment design:
(1)Authors claim this assay can detect COII gene in bovine blood plasma and serum samples. So how about the meat cell culture sample?
(2)In the present paper, the PCR analysis is able to detect bovine DNA. How about the specificity and sensitivity?
(3)The only PCR result is nowhere near enough to accurately analyze the DNA in sample.

Reviewer 2 Report

General Comment: The matter is under the scope of the journal and the manuscript has some interesting data.

-Abstract

Indicate the DNA extraction method and detection (primers used).

Please reduce the number of keywords

- Introduction:

I suggest the authors add recent research related to the topic of the manuscript. Please also add methods for the identification of animal species, meat adulteration and detection techniques.

Describe more clearly the process of perfect transformation and their use in meat products

- Materials and method

Indicate the methodology for the preparation of bovine serum and bovine blood plasma.

Analysis of cultured meat serum was not observed, nor analysis of fetal bovine serum (FBS),

Please indicate the statistical analysis applied

-Results

why the intensity of bands in blood plasma was greater than serum? compared to other studies. Please explain

L 145-158, this paragraph must be in the introduction

Please, from the results obtained, explain how the serum does not undergo a perfect transformation.

Reviewer 3 Report

The manuscript Foods-1916354 entitled “Species-Specifics Deoxyribonucleic Acid (DNA) Identification 2 of Bovine in Cultured Meat Serum for Halal Status” aimed to determine the halal status of cultured meat by detecting species-specific DNA of bovine serum as one of the medium used during meat production.

The paper generally reads well. The theme and the methodological approach are not novel. The authors did not design unique oligonucleotide primers, especially for their study, but they have used already designed primers in previous work and this will reduce the work novelty. The methods part lacks details of sample collection that is an essential part in the methods. The authors should clarify how many samples they collected and from where they collected the samples and also how many brands?

Line 129-131: Please rewrite the PCR program conditions.

Round 2

Reviewer 1 Report

According to the author’s reply, the paper is nowhere enough to publish in Foods.